# Thermally Conductive Styrene-Butadiene Rubber/Boron Nitride Nanotubes Composites

**Cristina S. Torres-Castillo and Jason R. Tavares ***

CREPEC, Chemical Engineering Department, Polytechnique Montreal, 2900 Edouard Montpetit Blvd,
Montréal, QC H3T 1J4, Canada
* Correspondence: jason.tavares@polymtl.ca

**Abstract:** The use of boron nitride nanotubes (BNNTs) for fabrication of thermally conductive composites has been explored in the last years. Their elevated thermal conductivity and high mechanical properties make them ideal candidates for reinforcement in polymeric matrices. However, due to their high tendency to agglomerate, a physical or chemical treatment is typically required for their successful incorporation into polymer matrices. Our previous study about the dispersibility of BNNTs allowed determination of good solvents for dispersion. Here, we performed a similar characterization on styrene-butadiene rubber (SBR) to determine its solubility parameters. Although these two materials possess different solubility parameters, it was possible to bridge this gap by employing a binary mixture. The solvent casting approach followed by hot pressing was chosen as a suitable method to obtain thermally conductive SBR/BNNT composites. The resulting nanocomposites showed up to 35% of improvement in thermal conductivity and a 235% increase in storage modulus in the frequency sweep, when a BNNT loading of 10 wt% was used. However, the viscoelastic properties in the amplitude sweep showed a negative effect with the increase in BNNT loading. A good balance in thermal conductivity and viscoelastic properties was obtained for the composite at a BNNT loading of 5 wt%.

**Keywords:** boron nitride nanotubes; styrene-butadiene rubber; thermal conductivity; nanocomposite

## 1. Introduction

Boron nitride nanotubes are ceramic nanoparticles with outstanding thermal conductivity, with theoretical values of up to 3000 W/(m·K) and experimental values around 350 W/(m·K) [1,2]. Like their carbon counterparts, the thermal conductivity varies according to the number of walls, with fewer walls corresponding to higher conductivity. In contrast with carbon nanotubes (CNTs), BNNTs are electrical insulators, which make them suitable for applications where thermal conductivity and electrical insulation are required, such as in devices for heat dissipation. BNNTs possess impressive mechanical properties (YounG's modulus of 1.2 TPa) [3], good chemical stability, and high thermal resistance, up to 900 °C in air [4]. Thermally conductive nanocomposites have been obtained with the incorporation of BNNTs and other BN nanostructures in polymeric matrices. Table 1 shows a summary of some previous studies.

**Table 1.** Summary of polymer/BN nanocomposites with improved thermal conductivity (TC).

| Matrix | BN Material | Aligned or Modified | Loading | TC Composite (W/(m·K)) | % TC Increase | Ref. |
|---|---|---|---|---|---|---|
| PS | | | 35 wt% | 3.61 | 1905 | |
| PMMA | BNNTs | No | 24 wt% | 3.16 | 2006 | [1] |
| PEVA | | | 37 wt% | 2.5 | 1370 | |
| PVB | | | 18 wt% | 1.81 | 654 | |
| PVF | BNNTs | No | 10 wt% | 0.45 | 150 | [5] |
| PVA | m-BNNTs | Yes | 3 wt% | ~0.30 | 267 | |

**Table 1.** *Cont.*

| Matrix | BN Material | Aligned or Modified | Loading | TC Composite (W/(m·K)) | % TC Increase | Ref. |
|---|---|---|---|---|---|---|
| PVA | (O)BNNTs | Yes | 10 wt% | 0.54 | 237 | [6] |
| Epoxy | BNNTs-BNNSs | Yes | 2 wt% | 0.47 | 147 | [7] |
| Epoxy | BNNTs | Yes | 30 wt% | 2.77 | 1285 | [8] |
| Epoxy | BNNTs | No | 30 wt% | 2.9 | 1350 | [9] |
| TPU | BNNTs | No | 1 wt% | 14.5 | ≥400 | [10] |
| PC | BN plates | Yes | 18.5 vol% | 3.09 | 115 | [11] |
| Epoxy | BN platelets | Yes | 50 wt% | 6.09 | 2800 | [12] |
| Epoxy resin | h-BN | Yes | 44 vol% | 9.0 | 4400 | [13] |
| Polysiloxane | BNNSs | Yes | 15 vol% | 1.56 | 290 | [14] |
|  | BN |  |  | 0.28 | 82 |  |
| SBR | BNNSs | Yes | 10. 5 vol% | 0.43 | 119 | [15] |
|  | Si-BNNSs |  |  | 0.57 | 253 |  |
|  | IBNNSs |  |  | 0.55 | 189 |  |
|  | (O)BNNSs |  |  | 1.08 | 468 |  |
| SBR | (R)PRh-BNNSs | Yes | 27.5 vol% | 0.75 | 295 | [16] |
|  | (O)PRh-BNNSs |  |  | 1.50 | 689 |  |

Si-BNNSs refers to silane-modified BNNSs; PRh-BNNSs represents polyrhodanine@BNNSs nanostructure; (R) and (O) stands for random and oriented nanostructures.

As can be seen, special treatments, alignment or surface modification of BN nanostructures are required to further increase thermal conductivity. These additional treatments imply not only longer processing times and subsequent purification steps, but also higher energy consumption, when the nanotubes have to be aligned by any means. Regarding SBR/BN composites, the incorporation of BN nanostructures has been limited to boron nitride nanosheets (BNNSs). This could be attributed to the more expensive methods required for the synthesis of high quality BNNTs. It was also observed that chemical functionalization with a silane agent [15] or coating with polyrhodanine [16] was needed to improve the compatibility of BNNSs with a rubber matrix. A more straightforward approach is missing for the fabrication of thermally conductive SBR/BN nanocomposites.

The incorporation of nanoparticles into polymers is often carried out in the liquid phase, by dispersing both materials in a suitable solvent. Thus, a solvent compatible with both the targeted polymer and nanomaterial should be used. The appropriate selection can be performed based on the solubility of the polymer and the filler.

Solubility parameters, also known as cohesion parameters, can be used to correlate the cohesion energies of polymers and liquids, by assessing the properties of individual compounds. Solubility parameters rely on the principle that "like seeks like", which means that a polymer will be dissolved in a liquid when the solubility parameters of both are alike [17]. The energy required to hold together the molecules of a material is known as the cohesive energy. This energy is divided into three components, accounting for dispersive, polar, and hydrogen bonding interactions (Equation (1)). Dividing each component by molar volume, we obtain the cohesive energy density. The square root of the cohesive energy density is known as the Hildebrand solubility parameter ($\delta t$), as can be seen in Equation (2):

$$\frac{E}{V} = \frac{E_d}{V} + \frac{E_p}{V} + \frac{E_h}{V}, \tag{1}$$

$$\delta_t = \sqrt{\delta_d{}^2 + \delta_p{}^2 + \delta_h{}^2} \tag{2}$$

where $\delta_d$, $\delta_p$, and $\delta_h$ are the Hansen solubility parameters. There are several ways to determine the Hansen solubility parameters (HSPs) of a polymer, such as dissolution, swelling, or viscosity changes [17,18]. The simplest way is to dissolve a certain amount of polymer in a specific volume of solvent. Then, after a period of time, the dissolution state is evaluated qualitatively by classifying the solutions in good or bad [17,19]. It is possible to represent the solubility space for the polymer and the solvents in a 3-D space (Hansen space). The center of the sphere is determined by the HSPs of the polymer, with coordinates $\{\delta_d; \delta_p; \delta_h\}$ and a calculated radius, $R_0$. The solvents located inside the sphere will dissolve

the polymer, while the solvents found outside will not. The distance between the polymer and the solvent can be estimated by applying Equations (3) and (4):

$$R_a{}^2 = 4(\delta_{d1} - \delta_{d2})^2 + (\delta_{p1} - \delta_{p2})^2 + (\delta_{h1} - \delta_{h2})^2 \qquad (3)$$

$$RED = \frac{R_a}{R_0} \qquad (4)$$

where $\delta_{d1}$, $\delta_{p1}$, and $\delta_{h1}$ correspond to the HSPs of the polymer and $\delta_{d2}$, $\delta_{p2}$, and $\delta_{h2}$ represent those of the solvent. The relative energy difference (*RED*) indicates where the solvent is located with respect to the polymer sphere. *RED* < 1 corresponds to good solvents, located inside the sphere, *RED* > 1 indicates that the solvent is located outside, and *RED* = 1 indicates that the solvent may be able to dissolve the particle to a certain degree [17].

In this work, the incorporation of pristine boron nitride nanotubes into styrene-butadiene rubber was performed through a simple approach. No surface modification or pre-treatment was needed. By analyzing the Hansen solubility parameters of both materials and pushing to the boundaries the solubility theory, a suitable dispersion medium is described. The thermal conductivities of the resulting nanocomposites are presented, along with the mechanical properties derived from rheological testing. The present work offers a facile technique for the incorporation of BNNTs into SBR for the fabrication of composites with improve thermal conductivity.

## 2. Materials and Methods

### 2.1. Materials

Purified BNNTs in the form of powder were provided by Tekna Plasma Systems. The average nanotube diameter was 5 nm ± 2nm, with a BNNT content of >75%. They were synthesized by an induction thermal plasma process (HABS method) [20], followed by purification with a thermal method [21]. Styrene-butadiene rubber (SBR) was provided by PT Synthetic Rubber Indonesia as a light brown solid product, comprising 27% polybutadiene (itself composed of 24% vinyl-1,2 units, 30% cis-1,4 units, and 46% trans-1,4 units). Its molecular weight Mn was 118,000 g/mol, and its Mooney viscosity ML(1 + 4) was 50 MU, with a glass transition temperature of −52 °C. Organic solvents were purchased from Sigma-Aldrich and Fisher Scientific. All were of high purity (99%), except d-Limonene (96%) and ethanol (95%). All the materials were used as received.

### 2.2. Methods

SBR dissolutions were prepared in different organic solvents with validated solubility parameters. The solvents included alkanes, ketones, amides, alcohols, a chloro-compound, esters, and aromatic compounds. These solvents were chosen to have dissolutions with different polarities. In brief, 500 mg of polymer were dissolved in 5 mL of the solvent. The dissolutions were left in a quiescent state for 24 h at room temperature. Then, the dissolution states were evaluated as good, intermediate, or bad, considering if the polymer was completely dissolved, partially dissolved or no dissolution was observed.

Dispersions of BNNTs were prepared in a Cole-Palmer sonicator operating at 20 kHz and at 30% of amplitude, with cycles of 5 s ON and 2 s OFF. The power delivered to the dispersions was ~25 W. A cylindrical probe (type 410-08) was employed; 10 mg, 50 mg, or 100 mg of BNNTs were dispersed in 10 mL of solvent. To determine the adequate amount of energy needed during sonication, preliminary tests were conducted at different energies. According to the guidelines provided by Girard et al. for the dispersion of cellulose nanocrystals [22], an energy of 167 kJ/g L (grams of the material; liters of the solvent) was recommended. Based on that, we calculated the energy required to disperse the BNNTs at three different loading: 10, 50, and 100 mg. The respective energies were 16.7 J, 83.5 J, and 167 J. However, when applying this energy to our system, we observed that the dispersions were not complete, and some agglomerates could still be seen. Thus, we increased the energy up to 500 J or 1000 J to have homogeneous dispersions. We concluded

that the energy applied during ultrasonication is not only a function of the amounts of the material and the volume used, but also of the type of nanoparticle to be dispersed and the properties of the solvent such as viscosity and surface tension. A total energy of 500 J (5 J/(mg·mL)) was applied to 10 mg of BNNT, and 1000 J were used for 50 mg (2 J/(mg·mL)) and 100 mg (1 J/(mg·mL)) of BNNTs.

SBR/BNNTs nanocomposites were fabricated in a two-step process. First, SBR dissolutions and BNNT dispersions were prepared at the desired concentrations (see Table 2). A fixed amount of SBR was dissolved in 10 mL of the retained solvent. The dissolution was left in a quiescent state overnight, to allow the polymer to be dissolved. To reach full dissolution, the system was magnetically stirred at 500 rpm for 4 h. While stirring, BNNT suspensions were prepared at the desired concentrations. Once the SBR and the BNNTs were fully dissolved or dispersed, they were mixed and stirred at 500 rpm for 1.5 h. The resulting mixtures were poured on 6 cm-diameter polytetrafluoroethylene (PTFE) dishes and allowed to dry at room temperature. Polymer films were obtained after 72 h of drying. The second step comprised the molding of the films. They were hot-pressed to evaporate any residual solvent and to eliminate possible bubbles/voids formed during the drying process. This was performed in a Carver press (model 3912) at a pressure of 3000 lbf/in$^2$ at 120 °C for 5 min, with a pre-heating step at 1200 lbf/in$^2$ for 2 min. Then, the discs were put in a room-temperature Carver press at 1200 lbf/in$^2$ for 3 min to cool down the samples. A schematic of this methodology is shown in Figure 1.

**Table 2.** Nomenclatures of composites prepared.

| BNNT Loading | Mass SBR | Mass BNNT |
|---|---|---|
| 0 wt% | 1 g | 0 g |
| 1 wt% | 0.990 g | 0.010 g |
| 5 wt% | 0.950 g | 0.050 g |
| 10 wt% | 0.900 g | 0.100 g |

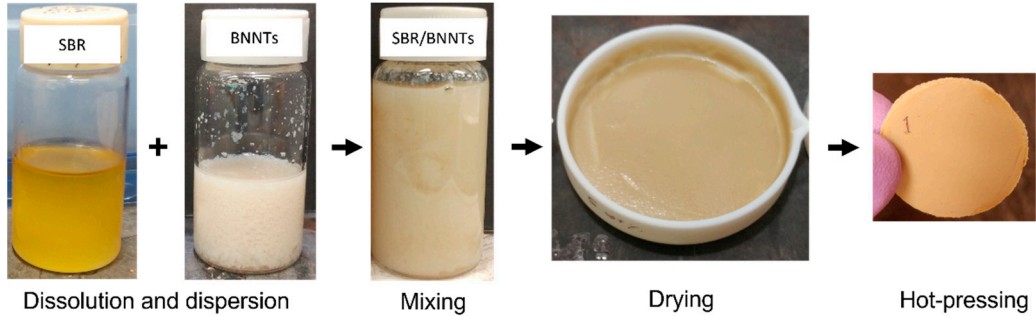

Dissolution and dispersion　　Mixing　　Drying　　Hot-pressing

**Figure 1.** Methodology followed for the fabrication of SBR/BNNTs nanocomposites.

Transmission electron microscopy (TEM) measurements were conducted in a Jeol JEM 2100F. Prior to the analyses, the samples were encapsulated in an epoxy resin and ultramicrotomed using an EM UC7 (Leica Microsystems, Wetzlar, Germany).

Rheological properties of the composites were measured in an Anton Paar rheometer, model MCR 502. Rough parallel plates (serial number: 18289) of 25 mm in diameter were used as the measurement system. Amplitude and frequency sweeps were performed at 23 °C. Amplitude sweeps were run at a frequency of 10 Hz (62.8 rad/s) in a strain range of 0.01–100%. Frequency sweeps were performed at a constant strain $\gamma$ of 0.1% in a frequency range of 0.1–100 rad/s.

Thermal conductivity measurements were performed at room temperature using a TCi Thermal Conductivity Analyzer (model TCi-3-A) from C-Therm. The modified transient plane source (MTPS) method was employed.

## 3. Results and Discussion

### 3.1. Hansen Solubility Parameters of Styrene Butadiene Rubber

HSP were determined for SBR. Three scores were given to the dissolutions, depending on whether the dissolution was complete (score "2"), partial (score "1") or no dissolution was observed (score "0") (see Figure S1). Table 3 shows the HSPs of each solvent, their scores, and the calculated Ra and RED numbers.

**Table 3.** HSPs, scores, and Ra and RED numbers of the solvents used for the dissolution of SBR.

| Solvent | $\delta_d$ | $\delta_p$ | $\delta_h$ | Score | $R_a$ | RED |
|---|---|---|---|---|---|---|
| d-Limonene | 17.2 | 1.8 | 4.3 | 2 | 0.81 | 0.16 |
| Chloroform | 17.8 | 3.1 | 5.7 | 2 | 1.64 | 0.33 |
| Toluene | 18 | 1.4 | 2 | 2 | 2.90 | 0.58 |
| Ethyl benzene | 17.8 | 0.6 | 1.4 | 2 | 3.64 | 0.73 |
| Ethyl benzoate | 17.9 | 6.2 | 6 | 2 | 4.15 | 0.83 |
| 1,4-Dioxane | 17.5 | 1.8 | 9 | 2 | 4.66 | 0.93 |
| Tetrahydrofuran | 16.8 | 5.7 | 8 | 2 | 4.96 | 0.99 |
| Cyclohexane | 16.8 | 0 | 0.2 | 2 | 5.03 | 1.01 |
| Ethyl acetate | 15.8 | 5.3 | 7.2 | 1 | 5.09 | 1.02 |
| Methyl ethyl ketone | 16 | 9 | 5.1 | 1 | 7.11 | 1.42 |
| Benzyl alcohol | 18.4 | 6.3 | 13.7 | 1 | 10.24 | 2.05 |
| Acetone | 15.5 | 10.4 | 7 | 0 | 9.14 | 1.83 |
| 2-Propanol | 15.8 | 6.1 | 16.4 | 0 | 12.93 | 2.59 |
| N,N'-Dimethylformamide | 17.4 | 13.7 | 11.3 | 0 | 13.15 | 2.63 |
| Dimethylsulfoxide | 18.4 | 16.4 | 10.2 | 0 | 15.19 | 3.04 |
| Propylene carbonate | 20 | 18 | 4.1 | 0 | 16.35 | 3.27 |
| Ethanol | 15.8 | 8.8 | 19.4 | 0 | 16.58 | 3.32 |
| Methanol | 15.1 | 12.3 | 22.3 | 0 | 20.92 | 4.18 |
| Ethylene glicol | 17 | 11 | 26 | 0 | 23.23 | 4.65 |
| Formamide | 17.2 | 26.2 | 19 | 0 | 27.84 | 5.57 |
| Tol/EA (20/80) [1] | 16.2 | 4.5 | 6.2 | 2 | 3.61 | 0.72 |

[1] Additional solvent used but not considered for the HSP determination of SBR. "Tol" stands for toluene, and "EA" stands for ethyl acetate.

Once the dissolutions were scored, we proceeded to input the information in the HSPiP software to calculate the HSPs of the SBR. The HSPs were {$\delta_d$; $\delta_p$; $\delta_h$} = {17.4; 2.5; 4.4} ± {0.4; 0.8; 0.6} MPa$^{1/2}$, with $\delta t$ of 18.1 and a radius *Ra* of 5.0 MPa$^{1/2}$. The FIT equation was 1, meaning that all the solvents that were ranked as good were inside the sphere and all the solvents ranked as bad were outside (see Figure 2).

Liu et al. [23] previously determined the HSPs of cured SBR through swelling experiments. The obtained values were {$\delta_d$; $\delta_p$; $\delta_h$} = {18.0; 2.9; 2.3} MPa$^{1/2}$, with a total solubility parameter, $\delta t \approx 18.4$ MPa$^{1/2}$, and *Ra* = 5.0 MPa$^{1/2}$. The SBR used was Buna VSL 4526-2, with a styrene content of 26 wt%, a vinyl content of 44.5 wt%, and a Mooney viscosity ML(1+4) of 5 MU. In their experiments, they observed that ethyl acetate and methyl ethyl ketone were located in a low swelling area, which agrees with our results, where they were considered as intermediate solvents to partially dissolve SBR. HSPs of SBR (brand Polysar) have been also reported in the HSP database [17,24]. The values are {17.6, 3.4, 2.7}, with *Ra* of 6.55 MPa$^{1/2}$. Although the same polymer was used in the determination of HSPs, there are some differences observed. Indeed, HSPs of polymers are sensitive to molecular weight and chemical composition. Thus, different compositions and/or proportions of the repeat units and molecular weight will lead to different HSP values [17].

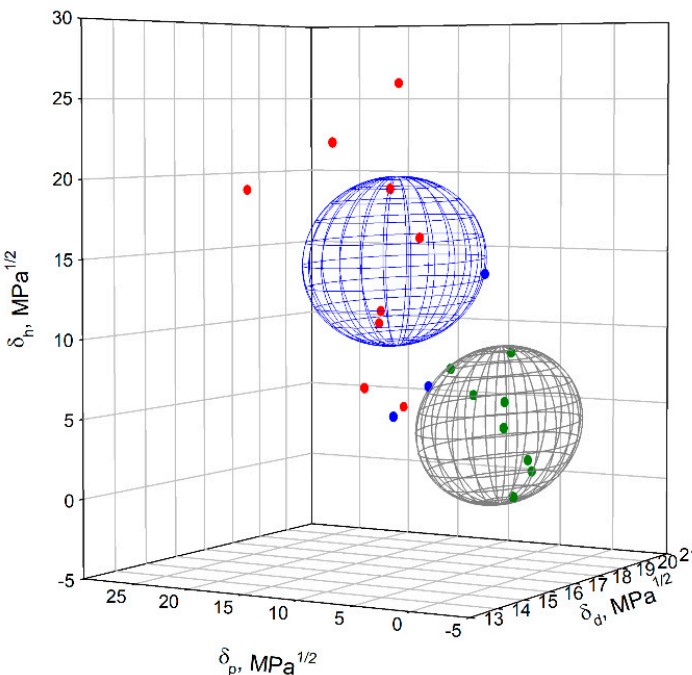

**Figure 2.** Hansen space for SBR (gray sphere) and BNNTs (blue sphere). Good solvents for SBR are represented with green dots, intermediate solvents are represented with blue dots, and bad solvents are represented with red dots.

As mentioned earlier, SBR and BNNTs have different surfaces energies and solubility parameters. The mild polarity of BNNTs and the non-polar behavior of SBR make them locate in different Hansen spaces (see Figure 2). However, based on the Hansen solubility theory, it is possible that two bad or intermediate solvents dissolve the material in question by using binary mixtures. By exploring this concept and calculating the Ra and RED numbers for binary mixtures, we were able to obtain full dissolution of SBR and good dispersion of BNNTs in short times. Based on our previous work on the dispersion of purified BNNTs [25], with HSP $\{\delta_d; \delta_p; \delta_h\}$ = {16.8; 10.7; 14.7} $\pm$ {0.3; 0.9; 0.3} MPa$^{1/2}$ and $Ra$ = of 5.4 MPa$^{1/2}$, we determined that the best solvents to disperse BNNTs were dimethyl formamide (DMF), dimethyl acetamide (DMAc), ethanol, and isopropanol. We also determined that some other solvents with lower polarity were able to disperse this nanomaterial for short periods, such as ethyl acetate. On the other hand, when determining the HSPs of SBR (this work), we observed that ethyl acetate was able to partially dissolve this polymer, with non-polar solvents such as toluene being preferred to have full dissolution. Taking into account this information, we prepared binary mixtures of toluene/ethyl acetate at different volume ratios (see Table 4). Based on the results, we observed that the mixture at a 20:80 volume ratio would ensure the full dissolution of SBR without compromising good BNNTs dispersion. The calculated RED number in relation to SBR was 0.71 (RED < 1), meaning that this mixture was inside of its Hansen sphere. In relationship to BNNTs, the RED number was 1.96. Although this value is slightly higher than 1 and the coordinate is located outside of the solubility sphere of BNNTs, the toluene/ethyl acetate mixture at a 20:80 vol% was able to form stable dispersions for short periods of time, sufficient to undertake solvent casting.

**Table 4.** Calculated HSPs and RED numbers of binary mixtures of toluene/ethyl acetate.

| Tol:EA Volume Ratio | $\delta_d$ | $\delta_p$ | $\delta_h$ | $R_a$ | RED |
|:---:|:---:|:---:|:---:|:---:|:---:|
| 0/1 | 15.8 | 5.3 | 7.2 | 5.09 | 1.02 |
| 0.1/0.9 | 16.0 | 4.9 | 6.7 | 4.32 | 0.86 |
| 0.2/0.8 | 16.2 | 4.5 | 6.2 | 3.54 | 0.71 |
| 0.3/0.7 | 16.5 | 4.1 | 5.6 | 2.78 | 0.56 |
| 0.4/0.6 | 16.7 | 3.7 | 5.1 | 2.03 | 0.41 |
| 0.5/0.5 | 16.9 | 3.4 | 4.6 | 1.33 | 0.27 |
| 0.6/0.4 | 17.1 | 3.0 | 4.1 | 0.79 | 0.16 |
| 0.7/0.3 | 17.3 | 2.6 | 3.6 | 0.85 | 0.17 |
| 0.8/0.2 | 17.6 | 2.2 | 3.0 | 1.43 | 0.29 |
| 0.9/0.1 | 17.8 | 1.8 | 2.5 | 2.15 | 0.43 |
| 1/0 | 18 | 1.4 | 2 | 2.90 | 0.58 |

*3.2. Fabrication Styrene-Butadiene Rubber/Boron Nitride Nanotubes Composites*

SBR/BNNTs composites were obtained by the solvent casting technique. SBR solutions and BNNT dispersions were prepared separately and combined. Table 2 shows the concentrations used and the nomenclature of the samples.

The obtained films (see Figure 2) had a thickness of ~300 μm and a diameter of 6.3 cm. With the increase in filler loading, a more uniform color in the films was observed. The neat SBR film had a light brown/orange color (see Figure 3a). With the increase in concentration, darker films with more uniform color were obtained (Figure 3b–d).

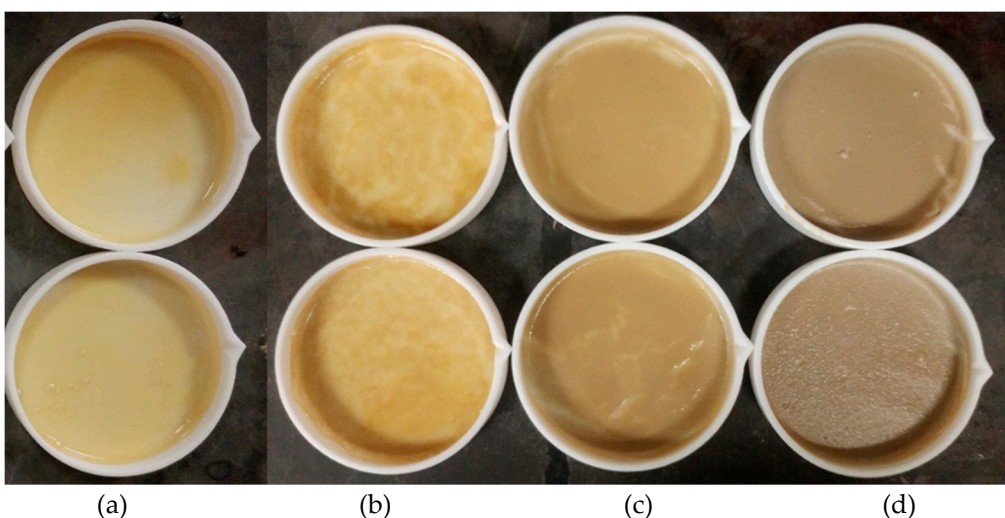

(a)　　　　　　(b)　　　　　　(c)　　　　　　(d)

**Figure 3.** SBR composites obtained by solvent casting containing 0 wt% (**a**), 1 wt% (**b**), 5 wt% (**c**), and 10 wt% (**d**) of BNNTs.

3.2.1. Rheology

Figure 4 shows the variation of the viscoelastic properties (G′ and G″) as a function of strain, obtained using a parallel-plate rheometer. The storage modulus (G′) curves are observed in Figure 4a. G′ of neat SBR and SBR/BNNT at 1 wt% overlapped in the linear viscoelastic (LVE) region, with an earlier decreased modulus at higher strains for the composite. The low BNNT loading did not allow the formation of a filler-filler network, and thus, the storage modulus was not affected. For the composites at BNNT contents of 5 wt% and 10 wt%, G′ was higher in the strain range of 0.01–1% but was reduced in the LVE region. This can be explained by the fact that at higher contents of BNNTs, a filler network is formed and, at higher strains, this network breaks. This phenomenon, characteristic of filled elastomers, is known as the Payne effect [26–29]. For unfilled rubber, the storage modulus has a linear behavior in almost of all the strain range. However, when a filler is

added, a filler-filler network is formed. A decreased G′ (or E′ for measurements performed in the tension mode) is observed at high strains, as a consequence of the rupture/disruption of the filler network [26–29]. Das et al. [29] prepared styrene-butadiene rubber-butadiene rubber blends (SBR-BR) reinforced with CNTs at different loadings. They observed the Payne effect in all the composites, with a more abrupt decrease in the storage modulus for the highest CNT concentration used. Similarly, Zhong et al. [30] observed the Payne effect on SBR−silica composites. The curves of loss modulus (G″) are appreciated in Figure 4b. For neat SBR and the SBR/BNNT composites at 1 wt% loading, a linear behavior occurred up to 1% strain. After this value, a small shoulder appeared. For the composites at BNNT loadings of 5 and 10 wt%, an increase in the G″ at low strains was observed. This time, the small shoulder appeared as a big peak at lower strains and with increased intensity. This behavior corroborated what is appreciated in the G′ curves: some disruptions of the filler-filler network occur at low strains [26]. However, due to the great flexibility provided by the rubber, the polymeric chains and the filler-filler network regenerate over time, as proven by Das et al. [29] in SBR-BR/CNTs composites. It has been widely accepted that the introduction of a filler in rubber leads to a drastic decrease in G′ and this is more evident with the increase in the loading of the filler. Accompanying this behavior, a peak in the G″ curve appears in the region where the G′ decreases. [26,28]. Figure 4c shows the complex viscosity (η*) curves of the composites. In all cases, a zero-shear viscosity (η$_0$) was obtained at low strains. η$_0$ increased with the BNNT loading, and it presented the highest value (9.7 kPa s) for the composite at 10 wt%. The linear region observed at low strains was reduced with the increase in the BNNT loadings. Shear-thinning behavior was observed for all the composites, with a decrease in η the in almost all the strain range. This is attributed to the rupture of the BNNT agglomerates (or the filler network) with strain. Shear-thinning has also been observed in carbon black suspensions [31].

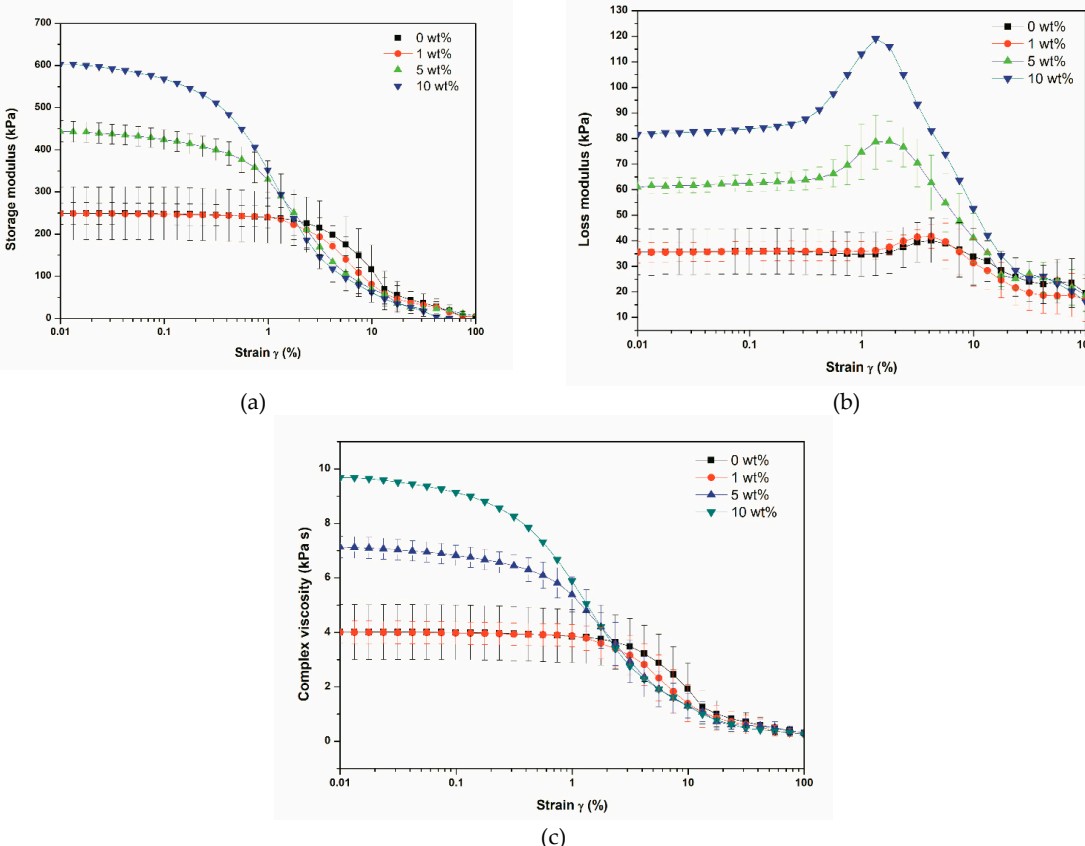

**Figure 4.** Variation of storage modulus (G′) (**a**), loss modulus (G″) (**b**), and complex viscosity (η*) (**c**) of SBR/BNNTs composites as a function of strain.

Figure 5 shows the G′, G″, and tan δ (corresponding to the G″/G′ ratio) curves of the composites as a function of angular frequency. The reinforcing effect of BNNTs was evident, with an improved storage modulus for all the composites in the whole frequency range (Figure 5a). The highest G′ was obtained for the composite at a loading of 10 wt% and at a frequency of 100 rad/s. The increase in the modulus was more pronounced at higher frequencies, with values of up to 246 kPa, 397 kPa, and 636 kPa for the composites at 1 wt%, 5 wt%, and10 wt%, respectively, at ω = 100 rad/s, in comparison to 190 kPa obtained for the unfilled rubber. Figure 5b shows the G″ curves for all the composites. Increases in the modulus in the composites were observed, which was related to higher energy dissipation. However, as can be seen in Figure 5c, the tan δ values (G″/G′ ratio) were lower for the composites. Although there was an increase in loss modulus with the increase in BNNT content, the elastic portion (G′) of the composites dominated in almost all the whole frequency range, with an overlapping of the curves at high frequencies. The decrease in tan δ curves of uncured SBR/carbon black composites has also been observed by Spanjaards et al. [31].

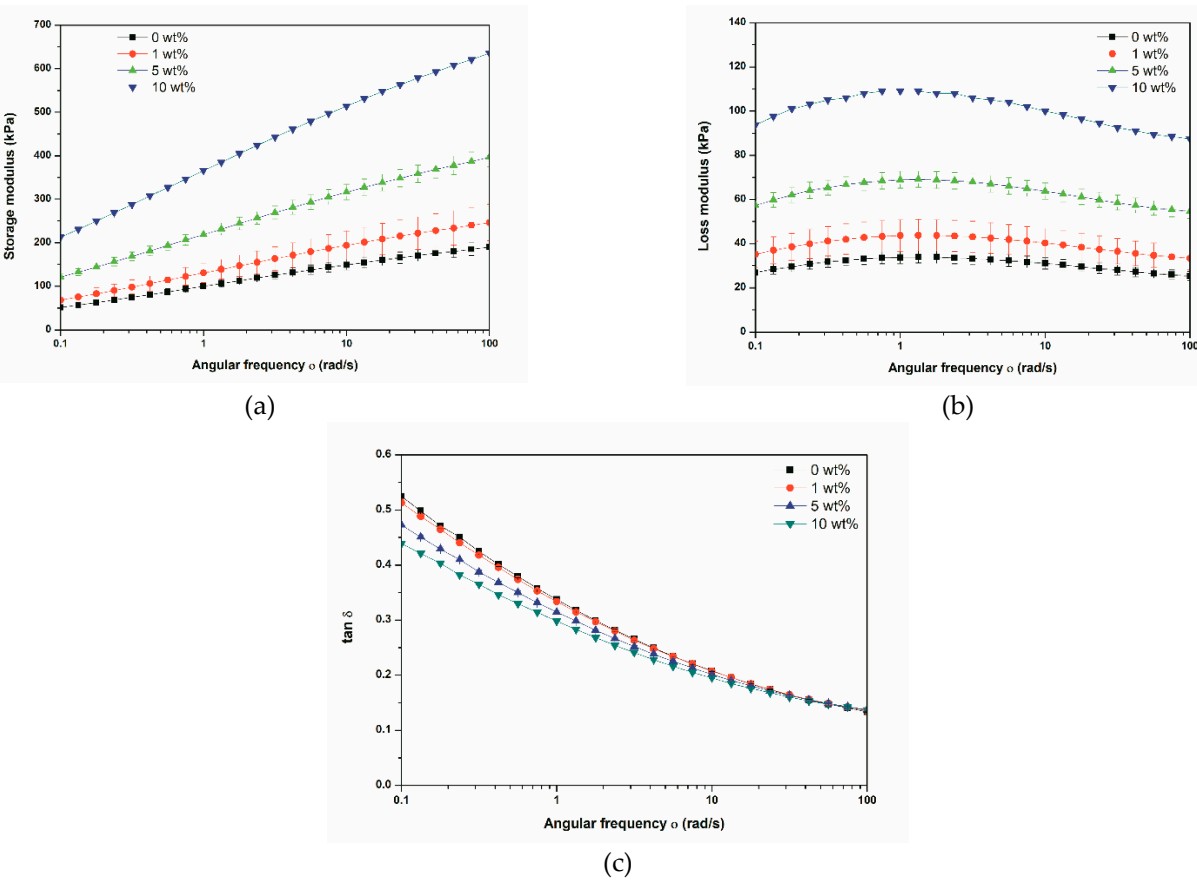

**Figure 5.** Variation of storage modulus (G′) (**a**), loss modulus (G″) (**b**), and tan δ (**c**) of SBR/BNNTs as a function of angular frequency.

### 3.2.2. Transmission Electron Microscopy

Figure 6 shows TEM images of the cross section of SBR/BNNTs composites at a filler loading of 1 wt%. Due to its very low thickness (~300 μm), the film was encapsulated in an epoxy resin and ultramicrotomed prior to analysis. The nanotubes were dispersed but randomly distributed in the film, and small agglomerates were formed in some regions, with absence of nanotubes in others. This concentration of the filler did not lead to the formation of a BNNTs network in the polymeric film, and thus, the nanotubes did not distribute homogeneously in the entire surface of the polymer film. This agrees with the

rheological results, where there was no effect in the viscoelastic properties (G′ and G″) in the amplitude sweep (Figure 4).

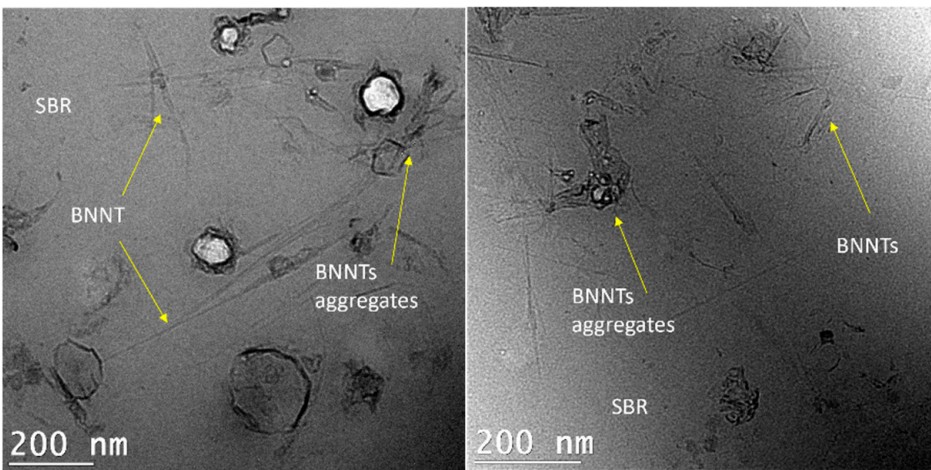

**Figure 6.** TEM images of SBR/BNNTs nanocomposites at 1 wt%.

### 3.2.3. Thermal Conductivity

Thermal conductivity results are shown in Figure 7. For the composites containing 1 wt% of BNNT (0.64 vol%), the thermal conductivity did not change with respect to the neat polymer. This agrees with the TEM observations (Figure 6), where it can be seen that, at this loading, the nanotubes were not homogeneously distributed in the SBR and, consequently, some regions without nanotubes were observed in the film. For the composites at 5 wt% and 10 wt%, thermal conductivities of 0.255 W/(m·K) and 0.268 W/(m·K), with increments of 29% and 35%, respectively, were observed, in comparison with pure SBR. As inferred from Figure 3c,d, the BNNTs were better distributed in the SBR matrix, forming a filler network and providing a uniform brown color along the composite. The formation of the filler network for these composites was corroborated in the rheological curves as a function of strain (Figure 4).

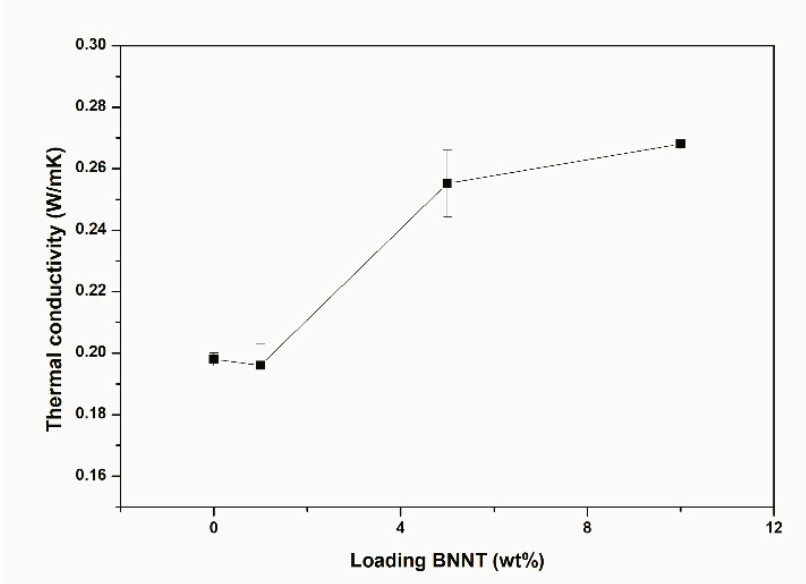

**Figure 7.** Thermal conductivity of the SBR/BNNTs nanocomposites.

Percolation thresholds of ~33 vol% and ~15 vol% for randomly oriented and aligned BNNTs, respectively, were determined for polyvinyl alcohol (PVA) composites [6]. Although we did not reach the percolation threshold in this work, we observed a significant

increment in thermal conductivity at loadings of 5 wt% (3.2 vol%) and 10 wt% (6.6 vol%). Further increases in BNNT loading and orientation of the nanotubes led to higher thermal conductivity in the rubber matrix. However, the strain-dependent viscoelastic properties (G′ and G″) were diminished based on the trends observed in Figure 4.

The alignment of BN nanomaterial in a polymer matrix plays an important role in thermal conductivity measurements. This alignment can be reached by applying magnetic or electric fields [6], shear forces [32], tape casting [33], and hot pressing techniques [11]. Terao et al. [6] prepared PVA/BNNT composites through solvent casting and electrospinning techniques. Low loadings of aligned nanotubes were needed to reach high thermal conductivities. Xie et al. [33] fabricated PVA/h-BN nanocomposites using the solvent casting approach. In their work, oriented h-BN microplatelets led to higher values in thermal conductivities, in comparison with randomly oriented BN. The tape casting method was responsible for the oriented platelets. Sun et al. [11] prepared polycarbonate/BN composites via hot pressing; a thermal conductivity of 3.09 W/(m·K) along the aligned direction was obtained for the composite at 18.5 vol%. Hu et al. [12] also applied the hot pressing technique for orientation of BN in epoxy resin. Their highest thermal conductivity (6.09 W/(m·K)) was obtained for oriented BN at a loading of 50%.

In this work, we did not intentionally align the nanotubes in the SBR. However, it has been reported that the solvent casting technique favors a partial alignment of the nanotubes in the polymeric matrix [33]. The hot pressing technique also has favored the alignment of the BN plates [1,11,12]. Thus, there might be a partial alignment in the nanotubes in our films. However, due to the moderate thermal conductivity values obtained, we believe that randomly oriented nanotubes were predominant in our composites.

Boron nitride nanomaterials have been incorporated into elastomers in order to fabricate thermally conductive composites. Cho et al. [14] fabricated poly(dimethylsiloxane) elastomer PDMS/BNNS composites. An increase of 23.5% in thermal conductivity was obtained at a BN loading of 15 vol%. Wu et al. [15] introduced BN or BNNS in SBR using the slurry approach on a conventional two-roll mill. Improvements of 82% and 119% were obtained for SBR/BN and SBR/BNNS, respectively, at a loading of 10.5 vol%. In a subsequent work, Wu et al. [16] coated BNNS with polyrhodanine to improve the compatibility with SBR. Composites with oriented nanosheets showed an improvement of up to 464% with respect to the neat matrix and a 187% improvement when randomly nanosheets were used. In both cases, the loading of BN nanostructures was 27.5 vol%.

Carbon-based nanofillers have also been used to reinforce SBR matrices. Das et al. [29] prepared SBR-BR/MWCNTs composites using a two-roll mill. A pre-dispersion of the nanotubes in ethanol was performed. An approximately 30% increase in thermal conductivity was obtained, when four parts per hundred of rubber (phr) of CNTs were used. Song et al. [34] fabricated SBR/reduced graphene oxide (RGO) composites. An improvement of 26.1% was obtained at a loading of 3 wt% of RGO. Further improvements were obtained when using functionalized CNTs (f-CNTs) and hybrid-RGO/f-CNTs as fillers. A modest increase in thermal conductivity for polymer/CNTs composites has been reported with increasing CNTs loading [35–37]. These modest values are attributed to the interfacial resistance between the polymer and CNTs [38]. However, it is often considered that there is not a true percolation threshold for these nanotubes, but this interpretation may vary, depending on the level of improvement. For example, Kwon et al. [39] determined a percolation threshold of 1.4 vol% for CNTs in PDMS/CNTs composites when a 390% increase in thermal conductivity was obtained. On the other hand, Kapadia et al. [38] did not report a percolation threshold even at higher CNTs loadings (10 vol%).

The thermal conductivity values obtained in this work are comparable to the ones reported in the literature when using SBR/carbon-based nanomaterials. Based on the Hansen solubility theory, we were able to make the non-polar SBR matrix compatible with the mildly polar behavior of BNNTs. We validated the Hansen solubility theory. The simple solvent casting approach developed in this work paves the route for the fabrication

of rubber composites with improved thermal and mechanical properties, while retaining electrical insulation (contrary to carbon nanotube systems).

## 4. Conclusions

SBR-BNNTs nanocomposites were fabricated by the solvent casting technique. Although both materials possess different surface energies, we proposed an intermediate media to mix them. A binary mixture of toluene/ethyl acetate in a 20:80 volume ratio was proposed as dispersion media, based on the Hansen solubility theory. This system allowed full dissolution of the polymer and good dispersion of the nanotubes. The nanocomposites at 10 wt% BNNT loading showed an increase in thermal conductivity of up to 35%, while the composites containing 5 wt% of BNNTs showed an increase of 29% in thermal conductivity. All the composites showed viscoelastic properties with an increase in loading of BNNTs in the frequency dependence curves. This work paves the way for subsequent incorporation of BNNTs into SBR, to eventually be followed by the vulcanization process.

**Supplementary Materials:** The following supporting information can be downloaded at: https://www.mdpi.com/article/10.3390/jcs6090272/s1, Figure S1: Dissolution of SBR in organic solvents after 24 h: (**a**) cyclohexane, methanol, toluene, acetone, methyl ethyl ketone, ethylene glycol, dimethyl sulfoxide, formamide, ethyl acetate, and ethanol; and (**b**) propylene carbonate, ethyl benzene, benzyl alcohol, dimethyl formamide, chloroform, tetrahydrofuran, ethyl benzoate, 2-Propanol, 1,4-Dioxane, and d-Limonene. Solvents were shown from left to right.

**Author Contributions:** Conceptualization, C.S.T.-C. and J.R.T.; methodology, C.S.T.-C.; software, C.S.T.-C.; validation, C.S.T.-C. and J.R.T.; formal analysis, C.S.T.-C.; investigation, C.S.T.-C.; resources, J.R.T.; data curation, C.S.T.-C. and J.R.T.; writing—original draft preparation, C.S.T.-C.; writing—review and editing, J.R.T.; supervision, J.R.T.; project administration, J.R.T.; funding acquisition, J.R.T. All authors have read and agreed to the published version of the manuscript.

**Funding:** Research was funded by the Natural Sciences and Engineering Research Council of Canada (NSERC) (CRDPJ 499340–2016); and Prima Quebec (R12-13-006). C.S.T.-C. would like to acknowledge the Mexican National Council for Science and Technology (CONACyT) for the scholarship provided (number: 739894).

**Institutional Review Board Statement:** Not applicable.

**Informed Consent Statement:** Not applicable.

**Data Availability Statement:** The data presented in this study are available in the Supplementary Information file, as well as on request from the corresponding author.

**Acknowledgments:** Thanks are given to Matthieu Gauthier for his assistance in the preparation of the samples for TEM. Special thanks go to Jeremy Mehlem for his guidance in the analysis of rheological data and to Gabriel Dion for his support in thermal conductivity measurements.

**Conflicts of Interest:** The authors declare no conflict of interest.

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
