# Peer review of "Thermally Conductive Styrene-Butadiene Rubber/Boron Nitride Nanotubes Composites"

_jcs, doi:10.3390/jcs6090272_

Round 1

Reviewer 1 Report

Dear authors,

I had the opportunity to review your work. Below some comments to improve its quality in view of a possible publication:

- to better describe all the experimental activities including also photo and other relevant data (setup, hardware, etc...);

- what is the optimal trend of percolation threshold according to CNT percentage?

- to extend the literature references as

Y. Liu, S. Kumar, Polymer/carbon nanotube nano composite fibers–a review, ACS applied materials & interfaces 6(9) (2014) 6069-6087.

P.-C. Ma, N.A. Siddiqui, G. Marom, J.-K. Kim, Dispersion and functionalization of carbon nanotubes for polymer-based nanocomposites: A review, Composites Part A: Applied Science and Manufacturing 41(10) (2010) 1345-1367.

 M. Arena, M. Viscardi, G. Barra, L. Vertuccio, L. Guadagno, Multifunctional performance of a Nano-Modified fiber reinforced composite aeronautical panel, 2019, Materials,16,6,869, doi: 10.3390/ma12060869.

and others.

- to add more details on the manufacturing part.

Reviewer 2 Report

It is necessary to make the following changes:

Remove duplication of data in Fig. 1,

remove duplication of data in transmission electron microscopy pictures

It is desirable to perform TEM studies for all the samples studied with different nanotube contents

Label all structural components on the TEM image

To improve the quality of Figure 6. Confidence intervals are not visible.

Reviewer 3 Report

This is well-written paper presenting useful and novel results. Some comments are given below.

1. Why were no error bars marked on the data points of the 10 wt% sample in Figures 3 and 4? Error bars were shown for others samples.

2. Similar condition to comment #1 was noticed in Figure 6. 

3. Authors indicated that orientation of the nanotubes would lead to higher thermal conductivity. However, no description of orientation of the nanotubes was given. It is suggested that more TEM images showing the samples with different BNNT loading percentages are required to clarify this issue.

4. Authors are suggested to explain in detail that which part of the results or figures were to show the viscoelastic properties of the composite. 

5. What is the purpose to increase the thermal conductivity of the rubber composite? The value obtained by this study is lower than that of the common rubber (about 0.5 W/mK). Authors may discuss any other potential additives.

Round 2

Reviewer 1 Report

Dear authors,

the paper is fine for me.

My regards.

Reviewer 2 Report

+

Reviewer 3 Report

The revised manuscript can now be accepted for publication.